# Recurrent Temporal Revision Graph Networks

**Yizhou Chen**[*]
Shopee Pte Ltd.,
Singapore

**Anxiang Zeng**[*]
SCSE, Nanyang Technological
University, Singapore

**Guangda Huzhang**[*]
Shopee Pte Ltd.,
Singapore

**Qingtao Yu**
Shopee Pte Ltd.,
Singapore

**Kerui Zhang**
Shopee Pte Ltd.,
Singapore

**Yuanpeng Cao**
Shopee Pte Ltd.,
Singapore

**Kangle Wu**
Shopee Pte Ltd.,
Singapore

**Han Yu**
SCSE, Nanyang Technological
University, Singapore

**Zhiming Zhou**[†]
Shanghai University of Finance
and Economics, China

## Abstract

Temporal graphs offer more accurate modeling of many real-world scenarios than static graphs. However, neighbor aggregation, a critical building block of graph networks, for temporal graphs, is currently straightforwardly extended from that of static graphs. It can be computationally expensive when involving all historical neighbors during such aggregation. In practice, typically only a subset of the most recent neighbors are involved. However, such subsampling leads to incomplete and biased neighbor information. To address this limitation, we propose a novel framework for temporal neighbor aggregation that uses the recurrent neural network with node-wise hidden states to integrate information from all historical neighbors for each node to acquire the complete neighbor information. We demonstrate the superior theoretical expressiveness of the proposed framework as well as its state-of-the-art performance in real-world applications. Notably, it achieves a significant $+9.6\%$ improvement on averaged precision in a real-world Ecommerce dataset over existing methods on 2-layer models.

## 1 Introduction

Many real-world applications (such as social networks [28, 41], recommender systems [3, 29], traffic forecasting [18, 45] and crime analysis [11]) involve dynamically changing entities and increasing interactions, which can be naturally modeled as temporal graphs (also known as dynamic graphs) where nodes and edges appear and evolve over time. Compared with static graph solutions [13, 35, 9], dynamic graphs can be more accurate in modeling node states and inferring future events.

In modern graph representation learning, neighbor aggregation, i.e., generating higher-level node embeddings via referring to the embeddings of their neighbors [13, 35, 17], is a critical common ingredient that helps more effectively leverage the connectivity information in the graphs. Practically, however, relatively hot nodes may have rapidly growing lists of neighbors, which makes involving all neighbors in neighbor aggregation computationally expensive. In static graphs, SAGE [8] suggests randomly sampling $d$ neighbors to approximate all neighbors. For temporal graphs, the prevalent approach is to consider only the most recent $d$ neighbors [28, 33]. But such subsampling of neighbors

---

[*]The three authors contributed equally to the paper.

[†]Corresponding author. zhouzhiming@mail.shufe.edu.cn.

37th Conference on Neural Information Processing Systems (NeurIPS 2023).

could lead to incomplete and biased information, as the neighbor information preceding the most recent $d$ neighbors is permanently lost.

To mitigate the problem caused by neighbor subsampling, we propose an alternative solution to classic neighbor aggregation for temporal graphs, called recurrent temporal revision (RTR), which can be used as a standard building block (a network layer) for any temporal graph network. Each RTR layer incorporates a hidden state for each node to integrate the information from all its neighbors using a recurrent neural network (RNN). The hidden state will serve as the embedding of each node at that layer, embracing the complete neighbor information and having the flexibility to focus more on long-term or short-term behaviors depending on the characteristics of the problem / dataset.

The existence of such a hidden state implies that all historical neighbors have already been integrated through previous updates. So when a new edge event comes, we may build a message with the latest neighbor to update the hidden state through the RNN. On the other hand, taking the state updates of previous neighbors into account can also be beneficial, which could provide additional information including new interactions of these neighbors [28] as well as updates in the neighbor states due to the influence of their own neighbors. In other words, re-inputting neighbors would enable revising their previously provided information. Therefore, we propose a module called temporal revision (TR) to collect state updates of previous neighbors to update the hidden state together with its latest interaction. To better capture the state updates, besides their current states, we additionally input the revisions and state changes of neighbors into the temporal revision module. Subsampling of such state updates of neighbors can be less problematic, since the state of each neighbor has already been observed and integrated by the hidden state in the past.

Towards further enhancement, we propose to introduce heterogeneity in temporal revision, i.e., treating involved nodes non-uniformly, to boost its theoretical expressiveness. Heterogeneity can be naturally incorporated, since when a node collects revisions from its neighbors, the neighbors may, the other way around, collect the revision from the node (at a lower level), which is, however, directly accessible to the node itself. We thus introduce heterogeneity by recursively identifying and marking such self-probing nodes as specialties in revision calculation. We theoretically show that such heterogeneous revision leads to improved expressiveness beyond Temporal-1WL [33], and practically verify its potential to promote effective information exchange among nodes.

Our contributions can be summarized as follows:

- We propose a novel neighbor aggregation framework for temporal graphs to tackle the challenges arising from the necessary subsampling of neighbors.

- We propose two expressiveness measures for temporal graph models in terms of temporal link prediction and temporal graph isomorphism test, and characterize their theoretical connection. We prove theoretically the superior expressiveness of our proposed framework, as well as provide several new results on the expressiveness of other baselines.

- We demonstrate in experiments that the proposed framework can significantly outperform state-of-the-art methods. In particular, we can consistently get improvements as the number of aggregation layers increases, while previous methods typically exhibit only marginal improvements [28, 38].

- An Ecommerce dataset is provided along with this submission, which can be valuable for evaluating temporal graph models in real-world settings.

## 2 Preliminaries

### 2.1 Temporal Graph Modeling

A temporal graph can be represented as an ordered list of timed events, which may include the addition, deletion, and update of nodes and edges. For simplicity, we will only discuss the handling of edge addition events in this paper, where newly involved nodes will be accordingly added and initialized. It can be easily extended to handle other events [28]. We denote the set of nodes and edges up to time $t$ as $V(t)$ and $E(t)$, respectively.

Modeling of temporal graph typically generates node embeddings that evolve over time as graph events occur, which can be used by downstream tasks such as link prediction [2, 29] and node classification [20, 4]. The base node embeddings can have various sources, including direct node

features [8, 35, 41], learnable parameters [9, 36], generated by other modules such as an RNN [28, 38, 33] or specifically designed module [3, 16, 22]). Modern temporal graph methods [28, 5, 33] typically employ additional layers on top of the base embeddings to (further) leverage the connectivity information within the graph, generating higher-level node embeddings with a wider receptive field that encompasses relevant edges and nodes.

## 2.2  (Temporal) Neighbor Aggregation

Among various approaches, neighbor aggregation is one of the commonly used techniques in various graph models as a critical ingredient, where the embedding of each node is updated by considering the embeddings of its neighbors. In the context of temporal graphs, the neighbor aggregation layer can be defined as follows [28, 38, 33]:

$$\mathbf{a}_u^k(t) = \mathsf{AGG}^k\Big(\big\{\big(\mathbf{h}_v^{k-1}(t), \Phi(e_{uv,t'})\big) \,\big|\, e_{uv,t'} \in E(t)\big\}\Big), \tag{1}$$

$$\mathbf{h}_u^k(t) = \mathsf{COMBINE}^k\Big(\mathbf{h}_u^{k-1}(t), \mathbf{a}_u^k(t)\Big), \tag{2}$$

where $t$ denotes the current time. $e_{uv,t'}$ denotes the event between node $u$ and node $v$ at time $t'(\leq t)$. $\Phi(e_{uv,t'})$ encodes the event feature, as well as its related time and sequence information, which may include the current time, the event time, the current number of interactions of $u$ and $v$, etc. $\mathsf{AGG}^k$ and $\mathsf{COMBINE}^k$ denote arbitrary learnable functions, which are typically shared at layer $k$. $\mathbf{a}_u^k(t)$ is the aggregated information from neighbors at the $k-1$ layer, while $\mathbf{h}_u^k(t)$ is the new embedding of node $u$ at layer $k$, combining its current embedding at the $k-1$ layer with the aggregated information $\mathbf{a}_u^k(t)$. This can be viewed as a straightforward extension of classic graph convolution layers [13, 35, 17] with additional time or sequence information.

## 2.3  Expressiveness of Temporal Graph Models

Expressiveness measure is crucial for identifying and comparing the capabilities of different models. While several studies have explored the expressiveness of (temporal) graph models [40, 5, 33, 39, 43], a widely accepted definition for the expressiveness of temporal graph models is currently lacking.

Towards addressing this gap and establishing a solid groundwork, we propose the following formal definition of expressiveness in terms of temporal graph isomorphism test and temporal link prediction:

**Definition 1.** *(Temporal graph isomorphism indistinguishability). Given a pair of temporal graphs $\langle G_1(t), G_2(t)\rangle$, we say they are indistinguishable w.r.t. a model $f$, if and only if there exists a bijective mapping between nodes of $G_1(t)$ and $G_2(t)$, such that, for each pair of nodes in the mapping, their embeddings generated by $f$ are identical at any time $t' \leq t$.*

**Definition 2.** *(Temporal link prediction indistinguishability). Given a temporal graph $G(t)$ and two pairs of nodes $\langle(u_1, v_1), (u_2, v_2)\rangle$ therein representing two temporal link prediction problems, we say they are indistinguishable w.r.t. a model $f$, if and only if the embedding of $u_1$ and $u_2$, $v_1$ and $v_2$ generated by $f$ are identical at any $t' \leq t$.*

**Definition 3.** *(Temporal graph isomorphism expressiveness). If a model $f_A$ can distinguish any pair of temporal graphs that model $f_B$ can distinguish, while being able to distinguish certain graph pair that $f_B$ fails, we say $f_A$ is strictly more expressive than $f_B$ in temporal graph isomorphism test.*

**Definition 4.** *(Temporal link prediction expressiveness). If a model $f_A$ can distinguish any pair of temporal link prediction problems that model $f_B$ can distinguish, while being able to distinguish certain pair of temporal link prediction problems that $f_B$ fails, we say $f_A$ is strictly more expressive than $f_B$ in temporal link prediction.*

We can similarly define equally expressive, more or equally expressive, and so on.

Through above, we have, for the first time, provided a consistent definition of expressiveness for both temporal graph isomorphism test and temporal link prediction. Besides, compared with existing definitions, ours is more concise and straightforward, avoiding the introduction of concepts such as the Identifiable Set as in [5] and Temporal Computation Tree as in [33].

Most existing results on the expressiveness of temporal graph models can be easily transferred to our definition. For example, message-passing temporal graph networks (MP-TGN) are strictly

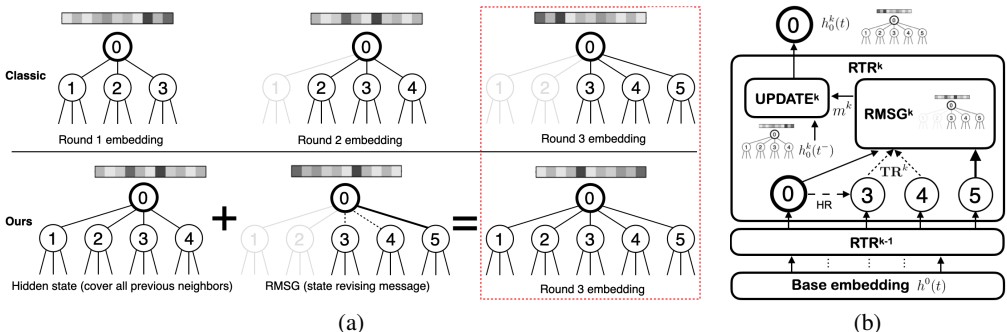

(a)                          (b)

Figure 1: (a) A comparison of our framework and other classic aggregations. Classic temporal aggregation typically involves subsampled neighbors in each step, which leads to incomplete and biased neighbor information, while our proposed RTR layer integrates the whole information of neighbors into node-wise hidden states. (b) An illustration of the inner structure of an RTR layer and our RTRGN, which builds $k$ successive RTR layers on top of the base embedding. We provide the pseudocode for RTRGN embedding calculation in Appendix A.

more expressive in isomorphism test when the (temporal) neighbor aggregation layers are injective (referred to as IMP-TGN) [33]; and IMP-TGN would be as expressive as temporal 1-Weisfeiler-Lehman (Temporal-1WL) test [33, 39], if with sufficient many layers.

## 3 The Proposed Method

### 3.1 Recurrent Temporal Revision

In large graphs, having too many neighbors can pose practical computation challenges in neighbor aggregation. For static graphs, SAGE [8] suggests approximating neighbor information by randomly subsampling a fixed number ($d$) of neighbors. For temporal graphs, recent models [28, 33] typically involve the most recent $d$ neighbors to reduce the computation cost of temporal neighbor aggregation, which often yields better performance compared with uniform sampling [28]. But such subsampling of neighbors fails to provide complete neighbor information and biases it towards short-term behaviors. The effects are illustrated in Figure 1(a).

Towards addressing this problem, we propose to introduce a hidden state $\mathbf{h}_u^k(t)$ in each aggregation layer $k > 0$ for each node to integrate information from all its historical neighbors under a state transition model, typically recurrent neural network (RNN). Since events in the temporal graph, and hence the addition of neighbors, occur sequentially, such integration can be done naturally.

Formally, when an event $e_{uv,t}$ occurs, we will need to update the hidden state $\mathbf{h}_u^k$ of $u$ (and also $v$). A straightforward implementation would be updating the hidden state with a message of the new event. However, taking the state updates of previous neighbors into account can also be beneficial, which can provide additional information including: the new interactions of neighbors, their state evolutionary directions, updates in the neighbor states due to the influence of their own neighbors. To achieve this, we introduce a module called temporal revision (TR) to collect state updates of previous neighbors, which will be later fused with the latest event to update the hidden state:

$$\mathbf{r}_u^k(t) = \mathsf{TR}^k\Big(\Big\{\big(\mathbf{r}_v^{k-1}(t), \Delta\mathbf{h}_v^{k-1}(t), \mathbf{h}_v^{k-1}(t), \Phi(e_{uv,t'})\big) \,\big|\, e_{uv,t'} \in E(t^-)\Big\}\Big), \qquad (3)$$

where $\mathsf{TR}^k$ is an arbitrary learnable function shared at layer $k$, $t^-$ represents the time right before $t$. We collect not only the newest states of neighbors $\mathbf{h}_v^{k-1}(t)$ at the $k-1$ layer, but also their state changes $\Delta\mathbf{h}_v^{k-1}(t) = \mathbf{h}_v^{k-1}(t) - \mathbf{h}_v^{k-1}(t^-)$ and their revisions $\mathbf{r}_v^{k-1}(t)$, as well as the event information $\Phi(e_{uv,t'})$, to help the model better extract the state update information. We set $\mathbf{r}_u^0(t)$ as zero vectors. When requesting the embedding of a node at the 0-th layer, we refer to its base embedding. Please refer to Appendix A for more detailed discussions of the base embedding.

In subsequent ablation studies, we show that retaining $\Delta\mathbf{h}_v^{k-1}(t)$ alone, while removing $\mathbf{h}_v^{k-1}(t)$, will not lead to degeneration of performance, which means the effectiveness in aggregation can rely

solely on the changes of states, while not necessarily on the neighbor states themselves. Temporal revision is hence substantially different from classic temporal aggregations [38, 28, 5, 33].

To encode all neighbor update information, we build a state revising message (RMSG) via:

$$\mathbf{m}_u^k(t) = \mathsf{RMSG}^k\big(\mathbf{h}_u^{k-1}(t), \mathbf{r}_u^k(t), \mathbf{h}_v^{k-1}(t), \Phi(e_{uv,t})\big), \tag{4}$$

where $\mathsf{RMSG}^k$ is an arbitrary learnable function shared at layer $k$, integrating the node embedding at the $k-1$ layer $\mathbf{h}_u^{k-1}(t)$, the embedding of the new neighbor at the $k-1$ layer $\mathbf{h}_u^{k-1}(t)$, the collected neighbor state update information $\mathbf{r}_u^k(t)$, as well as the event information $\Phi(e_{uv,t})$.[3]

Then the proposed recurrent temporal revision (RTR) layer can be formulated as:

$$\mathbf{h}_u^k(t) = \mathsf{UPDATE}^k\big(\mathbf{h}_u^k(t^-), \mathbf{m}_u^k(t)\big), \tag{5}$$

where the state revising message is consumed to update the hidden state $\mathbf{h}_u^k$ from time $t^-$ to $t$ via $\mathsf{UPDATE}^k$, a learnable function shared at layer $k$ responsible for hidden state transition. For all RTR layers, we initialize $\mathbf{h}_u^k(0)$ as zero vectors for all nodes.

## 3.2 Heterogeneous Revision

Heterogeneous message passing can improve the expressiveness of static GNN [43], which can be naturally integrated into our revision computation.

The term $\mathbf{r}_u^k(t)$ recursively gathers revision from previously interacted nodes, and they may in turn gather the revision of node $u$ (at a lower level), which is, however, directly accessible to $u$ itself. Therefore, we may adopt a different treatment for such self-probing calculations during recursive aggregation. To achieve this, we may mark all nodes in the recursive computation path as specialties.

Formally, we define the heterogeneous revision as:

$$\mathbf{r}_u^k(t, \mathcal{S}) = \mathsf{TR}^k\Big(\Big\{\big(\mathbf{r}_v^{k-1}(t, \mathcal{S} \cup \{u\}), \Delta\mathbf{h}_v^{k-1}(t), \mathbf{h}_v^{k-1}(t), \Phi(e_{uv,t'}), \mathbb{1}[v \in \mathcal{S}]\big) \,\Big|\, e_{uv,t'} \in E(t^-)\Big\}\Big), \tag{6}$$

where we introduce an extra argument $\mathcal{S}$ for revision, which is a set that is initialized as empty $\varnothing$ and recursively adds each $u$ encountered along the computation path. $\mathbb{1}$ stands for the indicator function, which triggers the difference between nodes in $\mathcal{S}$ and the rest, bringing about heterogeneity in the revisions. To switch to heterogeneous revision, we replace $\mathbf{r}_u^k(t)$ in (4) with $\mathbf{r}_u^k(t, \varnothing)$. We set the base case $\mathbf{r}_u^0(t, \mathcal{S})$ as zero vectors for any $u$ and $\mathcal{S}$.

We show in the next section that incorporating such heterogeneous revision theoretically leads to enhanced expressiveness, which will be subsequently validated in our later experiments.

We illustrate the complete inner structure of an RTR layer in Figure 1(b). The implementation details, including the design of all these learnable functions, are provided in Appendix A.

## 3.3 Model Expressiveness

For model expressiveness, we assume our RTR layers are built on top of a base RNN. For a fair comparison, it is assumed the base RNNs for all models are the same and of the best expressiveness, as suggested in [5, 33]. We focus on RTR layers whose (learnable) functions in (4) - (6) are injective, which can be achieved by adopting similar approaches as in [33, 39]. We refer to such a model as RTRGN. Besides, we assume node and edge attributes come from a finite set as [25, 5, 33].

We will mainly compare RTRGN with PINT [33], which is an instance of IMP-TGN that can be equally expressive as Temporal 1-WL in isomorphism test. PINT also offers augmented positional features (referred to as PINT-pos) designed to enhance its expressiveness.

Our first theoretical result demonstrates that the temporal graph isomorphism test is inherently more challenging than temporal link prediction. All proofs are deferred to Appendix B.

**Proposition 1.** *Being more expressive in temporal graph isomorphism test implies being more expressive in temporal link prediction ($f_A \succ_{is} f_B \Rightarrow f_A \succ_{lp} f_B$) but not vice versa ($f_A \succ_{lp} f_B \nRightarrow f_A \succ_{is} f_B$).*

---

[3]When there is no event at time $t$ for $u$, but requesting the message, we set $\mathbf{h}_v^{k-1}(t) = \mathbf{0}, \Phi(e_{uv,t}) = \mathbf{0}$ in (4).

Table 1: Summary of our theoretical results. Temporal graph isomorphism test is inherently more challenging than temporal link prediction. While Time-then-IMP and PINT are equally expressive as Temporal-1WL, our proposed RTRGN is strictly more expressive than the Temporal-1WL in both tasks. Relevant existing results are represented in gray for better reference.

| Expressiveness (**Link Prediction**) | Expressiveness (**Isomorphism Test**) |
|---|---|
| $f_A >_{is} f_B \Rightarrow f_A >_{lp} f_B$ (Prop. 1) | $f_A \cong_{is} f_B \Rightarrow f_A \cong_{lp} f_B$ (Corollary 1) |
| PINT-pos $>_{lp}$ PINT $>_{lp}$ MP-TGNs [33] | PINT $\cong_{is}$ Temporal-1WL [33] |
| Time-then-IMP $\cong_{lp}$ PINT $\cong_{lp}$ Temporal-1WL (Prop. 2) | Time-then-IMP $\cong_{is}$ PINT $\cong_{is}$ Temporal-1WL (Prop. 2) |
| RTRGN $>_{lp}$ PINT (Prop. 3) | RTRGN $>_{is}$ PINT (Prop. 3) |
| PINT-pos can be false positive (Prop. 4) | PINT-pos can be false positive (Prop. 4). |
| Most expressive: RNN + Temporal Relational Pooling (Prop. 5 in Appendix B) | |

The first part of this proposition can be proved by reducing the temporal link prediction problem into a temporal graph isomorphism test of the current graph adding the two temporal links separately. The second part of this proposition is proved by constructing a pair of two models: one is more expressive than the other in temporal link prediction, while being not more expressive in temporal graph isomorphism test. The proof and the construction of the counter-examples inside suggest that temporal graph isomorphism test is a more challenging task than temporal link prediction, as it requires global distinguishability rather than just local distinguishability of the involved nodes.

Next, we demonstrate that Time-then-IMP, the strongest among the three classes of temporal graph models suggested in [5] with the graph module of its time-then-graph scheme being injective message passing (IMP) to ensure 1WL expressiveness [39], is equally expressive as PINT and Temporal-1WL in both temporal graph isomorphism test and temporal link prediction.

**Proposition 2.** *Time-then-IMP is equally expressive as PINT and Temporal-1WL in both temporal graph isomorphism test and temporal link prediction.*

The proof establishes that all these models are equally expressive as Temporal-1WL in both temporal link prediction and temporal graph isomorphism test. The proposition suggests that Time-then-IMP, although utilizes a distinct form of base RNN to encode all historical features of nodes and edges (which comes at the cost of significantly more space needed to embed each edge), its expressiveness[4] is ultimately equivalent to the Temporal-1WL class, which includes models like PINT.

We next show that RTRGN is more expressive than PINT, as well as the Temporal-1WL class:

**Proposition 3.** *RTRGN is strictly more expressive than the Temporal-1WL class represented by PINT in both temporal graph isomorphism test and temporal link prediction.*

As we have shown, the expressiveness of all the previously mentioned models is bounded by the Temporal-1WL, which, although powerful, can fail in certain cases. To illustrate this, we construct a temporal graph isomorphism test task called Oscillating CSL (Figure 2) where all Temporal-1WL models, including Time-then-IMP, fail. The difficulty of the task stems from the regular structure of the graph snapshots and stacked graphs at each timestamp, which cannot be distinguished by Temporal-1WL tests. RTRGN excels in this task due to the heterogeneity introduced in its revision computation, which enables its expressiveness to go beyond the limit of Temporal-1WL.

**Proposition 4.** *PINT-pos can be false positive in both temporal graph isomorphism test and temporal link prediction, e.g., may incorrectly classify isomorphic temporal graphs as non-isomorphic.*

On the other hand, we observe that PINT-pos can produce false positives (unlike the other models). Specifically, there are cases where PINT-pos mistakenly categorizes isomorphic temporal graphs as non-isomorphic, and incorrectly distinguishes two temporal links that should not be distinguishable. It fails essentially because its positional feature is permutation-sensitive, while all other mentioned models are permutation-invariant, which is a desirable property in graph model design [5, 25]. This further highlights the superiority of heterogeneous revision over augmented positional features.

Finally, we formulate a most expressive model that can always output the ground truth (Appendix B).

---

[4]Note that, in our definition for both temporal link prediction and temporal graph isomorphism test, only node embeddings are considered. Time-then-IMP builds edge features. However, such edge features still do not contribute to distinguishing, for instance, between the two Oscillating CSL graphs in Figure 2 under temporal isomorphism test. This is because, at any given time, the set of neighbors for any node in either of the two graphs remains identical: two neighbors with a consistently connected edge, two with the edge showing a pattern of occurrence $[1, 0, 1, 0, \dots]$, and two with a pattern of occurrence $[0, 1, 0, 1, \dots]$.

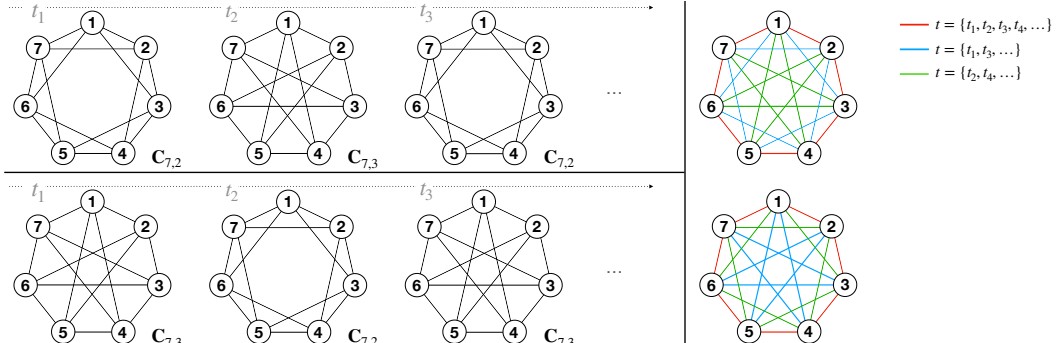

Figure 2: A synthetic temporal graph isomorphism test task called Oscillating CSL, where only RTRGN is expressive enough to give the correct result. The task is to distinguish the top and bottom temporal graphs, both of which oscillate but in different order between two regular structures called circular skip links (CSL). The right side shows the stacked view of the two temporal graphs.

## 4 Related Works

Early approaches in dynamic graphs learning focused on discrete-time dynamic graphs (DTDGs), which consist of a sequence of graph snapshots usually sampled at regular intervals [27, 6, 42]. They employed techniques such as stacking graph snapshots and then applying static methods [19, 1, 10, 31] and assembling snapshots into tensors for factorization [44, 23]. Among them, ROLAND [42] introduced hierarchical hidden states, but each state update is followed by a heavy finetuning of the entire model, which can be computationally expensive when applied to continuous-time dynamic graphs (CTDGs). In comparison, our framework is directly applied to CTDGs, handling the lists of timed events, and can be much more efficient.

DTDGs can always be converted to CTDGs without information loss. However, the high computation cost poses a major obstacle for applying most DTDG methods to CTDGs. More and more research focuses on CTDGs. Sequence-based methods such as DeepCoevolve [3], Jodie [16], and Dyrep [34] utilize RNN to capture the evolution of node representations, while some other works [26, 24] employ random walks to model CTDG, including CAWN [37]. NAT [21] caches the high-order neighbor information for computing joint neighbor features.

TGAT [38] and TGN [28] were the first to introduce GNN for CTDGs. Subsequently, combining GNNs with RNNs for temporal graph embedding generation became a mainstream idea. PINT [33] introduced MLP-based injective aggregators to replace attention-based aggregators in TGN. GRU-GCN [5] analyzed three strategies to combine GNN with RNN and classified them into time-and-graph (e.g., [30, 17, 2]), time-then-graph (e.g., [38, 28]), and graph-then-time frameworks (e.g., [27, 7, 29]).

Our heterogeneous revision leads to enhanced expressiveness. Similar ideas were adopted in static GNNs. For example, [43] introduced heterogeneity in the root node during message passing.

## 5 Experiments

### 5.1 Temporal Link Prediction on Benchmark Datasets

We evaluate the proposed RTRGN with the temporal link prediction task on 8 real-world datasets (7 public datasets as well as a private Ecommerce dataset that will be released along with this submission), and compare it against related baselines, including Dyrep [34], Jodie [16], TGAT [38], TGN [28], CAWN [37], NAT [21], GRU-GCN [5] and PINT [33]. The evaluation is conducted in two settings: the transductive which predicts links involving nodes observed during training, and the inductive which predicts links involving nodes that are never observed during training.

The official implementations of different baselines can vary in their experiment settings, especially on the *dataset preparation* and the *negative edge sampling*. To ensure a fair and consistent comparison, we calibrate the experiment settings of all baselines with the one used by TGN, which is the same as TGAT and PINT, but differs somewhat from other works such as CAWN and NAT. The latter may

Table 2: Average Precision (AP, %) on temporal link prediction tasks with 1-layer models ($k = 1$). First and second best-performing methods are highlighted. Results are averaged over 10 runs.

| | Model | MovieLens | Wikipedia | Reddit | SocialE.-1m | SocialE. | UCI-FSN | Ubuntu | Ecommerce |
|---|---|---|---|---|---|---|---|---|---|
| Transductive | Dyrep | 78.07 ± 0.2 | 95.81 ± 0.2 | 98.00 ± 0.2 | 81.74 ± 0.4 | 88.95 ± 0.3 | 53.67 ± 2.1 | 84.99 ± 0.4 | 62.98 ± 0.3 |
| | Jodie | 78.42 ± 0.4 | 96.15 ± 0.4 | 97.29 ± 0.1 | 70.20 ± 1.8 | 81.83 ± 1.1 | 86.67 ± 0.7 | 91.32 ± 0.2 | 68.70 ± 0.6 |
| | TGAT | 66.64 ± 0.5 | 95.45 ± 0.1 | 98.26 ± 0.2 | 51.97 ± 0.6 | 50.75 ± 0.2 | 73.01 ± 0.6 | 81.69 ± 0.3 | 65.13 ± 0.5 |
| | TGN | 84.27 ± 0.5 | 97.58 ± 0.2 | 98.30 ± 0.2 | 90.01 ± 0.3 | 91.06 ± 1.5 | 86.58 ± 0.3 | 90.18 ± 0.1 | 83.69 ± 0.5 |
| | CAWN | 82.10 ± 0.4 | 98.28 ± 0.2 | 97.95 ± 0.2 | 84.62 ± 0.4 | 83.49 ± 0.7 | 90.03 ± 0.4 | - | 84.58 ± 0.1 |
| | NAT | 75.85 ± 2.6 | 98.27 ± 0.1 | 98.71 ± 0.2 | 85.84 ± 0.4 | 91.35 ± 0.6 | 92.80 ± 0.4 | 90.86 ± 0.5 | 81.69 ± 0.2 |
| | GRU-GCN | 84.70 ± 0.2 | 96.27 ± 0.2 | 98.09 ± 0.3 | 86.02 ± 1.2 | 84.22 ± 2.5 | 90.51 ± 0.4 | 86.52 ± 0.2 | 79.52 ± 0.4 |
| | PINT | 83.25 ± 0.9 | 98.45 ± 0.1 | 98.39 ± 0.1 | 80.97 ± 2.2 | 84.94 ± 3.4 | 92.68 ± 0.5 | 89.59 ± 0.1 | 80.36 ± 0.3 |
| | RTRGN (ours) | 86.56 ± 0.2 | 98.56 ± 0.2 | 99.00 ± 0.1 | 92.20 ± 0.1 | 94.02 ± 0.3 | 96.43 ± 0.1 | 96.69 ± 0.1 | 88.05 ± 0.4 |
| Inductive | Dyrep | 74.47 ± 0.3 | 94.72 ± 0.2 | 97.04 ± 0.3 | 75.58 ± 2.1 | 88.49 ± 0.6 | 50.43 ± 1.2 | 71.49 ± 0.4 | 53.59 ± 0.7 |
| | Jodie | 74.61 ± 0.3 | 95.58 ± 0.4 | 95.96 ± 0.3 | 73.32 ± 1.4 | 79.58 ± 0.8 | 71.23 ± 0.8 | 83.81 ± 0.3 | 61.69 ± 0.6 |
| | TGAT | 66.33 ± 0.4 | 93.82 ± 0.3 | 96.42 ± 0.3 | 52.17 ± 0.5 | 50.63 ± 0.1 | 66.89 ± 0.4 | 78.92 ± 0.5 | 64.58 ± 0.8 |
| | TGN | 82.07 ± 0.2 | 97.05 ± 0.2 | 96.87 ± 0.2 | 88.70 ± 0.5 | 89.06 ± 0.7 | 81.53 ± 0.2 | 81.81 ± 0.4 | 81.81 ± 0.5 |
| | CAWN | 74.50 ± 0.5 | 97.70 ± 0.2 | 97.37 ± 0.3 | 75.39 ± 0.4 | 81.55 ± 0.5 | 89.65 ± 0.4 | - | 83.30 ± 0.4 |
| | NAT | 77.56 ± 0.6 | 97.74 ± 0.4 | 97.19 ± 0.7 | 85.16 ± 1.2 | 85.32 ± 3.6 | 87.83 ± 0.5 | 81.69 ± 0.9 | 76.85 ± 0.2 |
| | GRU-GCN | 82.81 ± 0.2 | 93.50 ± 0.2 | 96.38 ± 0.3 | 83.76 ± 1.8 | 79.21 ± 4.5 | 85.45 ± 0.6 | 73.71 ± 0.5 | 78.54 ± 0.2 |
| | PINT | 81.49 ± 0.6 | 97.29 ± 0.1 | 97.69 ± 0.1 | 77.45 ± 1.9 | 71.86 ± 3.6 | 90.25 ± 0.3 | 85.74 ± 0.2 | 77.15 ± 0.5 |
| | RTRGN (ours) | 85.03 ± 0.2 | 98.06 ± 0.2 | 98.26 ± 0.1 | 91.73 ± 0.7 | 92.47 ± 0.3 | 94.26 ± 0.1 | 92.95 ± 0.1 | 86.91 ± 0.5 |

Table 3: Average Precision (%) on temporal link prediction tasks with 2-layer models ($k = 2$).

| | Model | MovieLens | Wikipedia | Reddit | SocialE.1m | SocialE. | UCI-FSN | Ubuntu | Ecommerce |
|---|---|---|---|---|---|---|---|---|---|
| Trans | TGN | 85.33 ± 0.5 | 98.38 ± 0.2 | 98.55 ± 0.1 | 91.69 ± 0.3 | 91.56 ± 0.4 | 86.69 ± 0.3 | 92.22 ± 0.3 | 85.59 ± 0.2 |
| | NAT | 80.04 ± 1.2 | 98.15 ± 0.1 | 98.85 ± 0.1 | 90.41 ± 0.3 | 91.68 ± 0.6 | 93.03 ± 0.1 | 90.40 ± 0.7 | 82.32 ± 0.8 |
| | PINT | 84.18 ± 0.2 | 98.33 ± 0.1 | 98.56 ± 0.1 | 83.55 ± 1.5 | 84.56 ± 1.2 | 93.48 ± 0.4 | 90.32 ± 0.1 | 82.35 ± 0.7 |
| | RTRGN | 94.90 ± 0.4 | 98.79 ± 0.2 | 98.89 ± 0.1 | 93.30 ± 0.1 | 94.62 ± 0.1 | 98.04 ± 0.1 | 98.43 ± 0.1 | 95.28 ± 0.1 |
| Ind | TGN | 83.43 ± 0.5 | 97.85 ± 0.2 | 97.27 ± 0.4 | 90.45 ± 0.5 | 89.29 ± 0.6 | 83.58 ± 0.3 | 85.28 ± 0.3 | 84.31 ± 0.2 |
| | NAT | 72.23 ± 1.6 | 97.73 ± 0.2 | 97.32 ± 0.1 | 90.87 ± 0.2 | 90.56 ± 0.9 | 88.11 ± 0.6 | 79.44 ± 0.8 | 78.72 ± 1.8 |
| | PINT | 81.95 ± 0.5 | 96.39 ± 0.1 | 96.59 ± 0.2 | 81.83 ± 1.7 | 79.79 ± 1.6 | 91.23 ± 0.3 | 87.32 ± 0.2 | 80.30 ± 1.0 |
| | RTRGN | 93.37 ± 0.3 | 98.28 ± 0.2 | 97.76 ± 0.4 | 93.31 ± 0.1 | 93.99 ± 0.2 | 95.62 ± 0.1 | 96.75 ± 0.1 | 92.87 ± 0.1 |

lead to inflated results, with inductive performance surpassing transductive performance, potentially reaching over 99% accuracy in CAWN and NAT. We set $d = 10$ for our experiments. More details about the datasets, the baselines, and the experiment settings can be found in Appendix C.

The results for the 1-layer setting ($k = 1$) are presented in Table 2. It is evident that our proposed RTRGN surpasses all baselines by a significant margin, suggesting a substantial improvement. To delve deeper into the problem, we further examine the results for the 2-layer setting (Table 3), where we compare RTRGN against baselines that have demonstrated potentially better performance than other baselines in the 1-layer setting. As we can see, the performances of the baselines show limited improvement when transitioning from the 1-layer to 2-layer setting, aligning with previous findings [38, 28]. By contrast, RTRGN consistently exhibits *significant* improvements in the 2-layer setting: the average gap of transductive AP between RTRGN and the second best method is +4.4%. The largest gap appears in our Ecommerce dataset, which is +9.6%. The gap on Wikipedia is small probably due to a 1-layer model is already sufficient to give strong performance.

This indicates that layer design is important for temporal graph networks to more effectively leverage the power of multiple layers. Another advantage, as shown in Figure 3, we found in experiments is that the training converges faster for larger $k$.

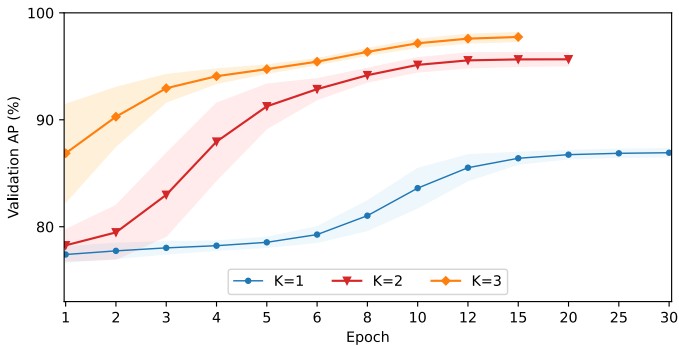

Figure 3: The convergence trend for different number of layers $k$.

## 5.2 Temporal Graph Isomorphism Test

To validate the superior expressiveness of the proposed RTRGN against existing methods, we conduct experiments of temporal graph isomorphism tests on two synthetic datasets. We run 4-layer models on testing graphs, where at each time step, we compute and store the embedding for each node. To check the isomorphism between two temporal graphs under a model, we gather the lists of generated embeddings (stacking in time for each node) of the two graphs and verify whether there exists a bijective mapping such that the two lists are identical (via sort and scan). If the two lists are identical, we conclude that the model outputs isomorphic (negative); otherwise, non-isomorphic (positive).

**Oscillating CSL** is a synthetic temporal graph isomorphism test task, which is a commonly used benchmark for assessing the expressiveness of GNNs [25, 5]. Each graph in this task is composed of a sequence of graph snapshots (of length 6) that correspond to the oscillation between 2 different prototypes randomly selected from $\{\mathbf{C}_{11,2}, \ldots, \mathbf{C}_{11,5}\}$. Here $\mathbf{C}_{N,s}$ denotes a circular skip link (CSL) graph with $N$ nodes and skip length $s$. The goal is to determine whether the temporal graph follows the pattern $\mathbf{C}_{N,s_1} \to \mathbf{C}_{N,s_2}$ is isomorphic to its reversed counterpart $\mathbf{C}_{N,s_2} \to \mathbf{C}_{N,s_1}$ (Figure 2).

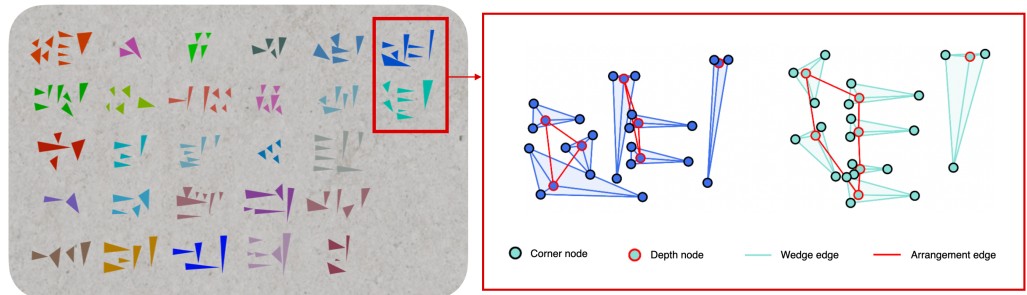

Figure 4: *(left)* An illustration of the Hittite cuneiform signs. *(right)* An example where TGN and GRU-GCN fail to output non-isomorphic, due to the special structure within the arrangement edges, i.e., two triangles versus a hexagon.

**Cuneiform Writing** is a modified version of the Cuneiform graph dataset [14] (Figure 4), which consists of graphs representing Hittite Cuneiform signs. The dataset was originally derived from tablets, where individual wedges along with their arrangements were automatically extracted. We draw out arrangement edges with short spatial distances, while ensuring that each depth node has at most 2 arrangement edges. To transform static graphs into temporal graphs, we adopt sequential edge generation that adds arrangement edges first, then the wedge edges. For each task, we randomly select 2 cuneiform signs with an equal number of nodes. Refer to Appendix C for more details.

Table 4: Isomorphism test results. Non-isomorphic is regarded as positive. TNR is true negative rate.

|  | Oscillating CSL | | | | | Cuneiform Writing | | | | |
|---|---|---|---|---|---|---|---|---|---|---|
|  | Precision | Recall | TNR | Accuracy | AUC | Precision | Recall | TNR | Accuracy | AUC |
| TGN | 0.00 | 0.00 | **100.0** | 16.67 | 50.0 | **100.0** | 74.19 | **100.0** | 79.80 | 86.98 |
| GRU-GCN | 0.00 | 0.00 | **100.0** | 16.67 | 50.0 | **100.0** | 95.44 | **100.0** | 96.43 | 97.61 |
| PINT-pos | 83.33 | **100.0** | 0.00 | 83.33 | 50.0 | 82.03 | **100.0** | 21.09 | 82.85 | 60.93 |
| RTRGN (w/o HR) | 0.00 | 0.00 | **100.0** | 16.67 | 50.0 | **100.0** | 95.44 | **100.0** | 96.43 | 97.61 |
| RTRGN | **100.0** | **100.0** | **100.0** | **100.0** | **100.0** | **100.0** | **100.0** | **100.0** | **100.0** | **100.0** |

The results are shown in Table 4. RTRGN achieves perfect accuracy in both experiments, attributed to its heterogeneous revision (HR). Notably, PINT-pos fails entirely to correctly identify true negatives in the Oscillating CSL task. TGN and GRU-GCN on the other hand exhibit partial failures in accurately distinguishing challenging non-isomorphic graphs. One such failure case is illustrated in Figure 4.

## 5.3 Ablation Studies

We conduct a set of ablation studies to dissect the impact of each proposed component in our RTRGN. The experiments are conducted on three representative datasets: MovieLens, Wikipedia, and UCI. The results for MovieLens are presented in Figure 5, while the results for the remaining two datasets, which validate the consistency of our findings, are included in Appendix D.

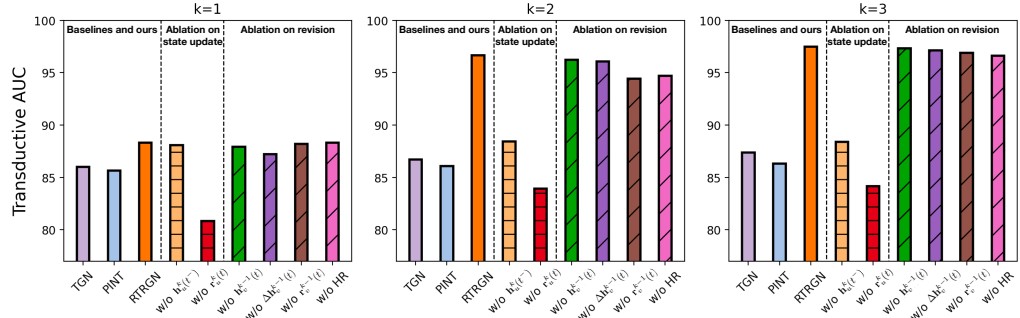

Figure 5: Transductive AUC (%) on MovieLens dataset. Results averaged over 10 runs.

**Ablation on state update.** Our first ablation study is on $\mathbf{h}_u^k(t^-)$, where we set $\mathbf{h}_u^k(t^-) = \mathbf{0}$ in (5) for $k > 0$, hence do not retain the hidden state but compute aggregation dynamically, denoted as w/o $\mathbf{h}_u^k(t^-)$. The results suggest that including the hidden state is crucial for further improving performance when $k > 1$. As the second ablation study, we set $\mathbf{r}_u^k(t) = \mathbf{0}$ in (4), which leads to a pure RNN design without neighbor aggregation, denoted as w/o $\mathbf{r}_u^k(t)$. It results in significant performance drops, indicating the criticality of integrating revisionary neighbor information in the message.

**Ablation on TR module.** The rest ablation studies are performed on the temporal revision module. As the first, we remove $\mathbf{h}_v^{k-1}(t)$ in (6), denoted as w/o $\mathbf{h}_v^{k-1}(t)$. Interestingly, this setting exhibits only a marginal performance drop and even slightly outperforms the setting denoted as w/o $\Delta\mathbf{h}_v^{k-1}(t)$ where $\Delta\mathbf{h}_v^{k-1}(t)$ is removed instead. It suggests that the effectiveness of revision can primarily rely on the state change of neighbors, not necessarily the neighbor states. This may be attributed to that the hidden states have already incorporated such information through previous messages. As the last two settings, we remove $\mathbf{r}_v^{k-1}(t)$ in (6), denoted as w/o $\mathbf{r}_v^{k-1}(t)$, and use the homogeneous version of revision (3) instead of the heterogeneous version (6), denoted as w/o HR. Both settings result in noticeable performance drops, indicating the benefits of incorporating revisions of neighbors in the TR module, as well as the introduction of heterogeneity in revision.

We provide the theoretical analysis of the time and space complexity of RTRGN in Appendix A. More experiments, e.g., ablation studies on subsampling amount $d$ and practical run-time comparisons, can be found in Appendix D.

## 6 Conclusion

We have proposed a new aggregation layer, named recurrent temporal revision (RTR) for temporal graphs. We have shown that it leads to superior expressiveness that goes beyond Temporal-1WL. We have also demonstrated that it can lead to significant improvements over current state-of-the-art methods. As an important breakthrough, we have demonstrated that it enables effective utilization of multiple aggregation layers, while existing methods have difficulties in effectively harnessing the power of multiple layers.

## 7 Acknowledgements

This research is supported in part by the National Natural Science Foundation of China (U22B2020); the National Research Foundation, Singapore and DSO National Laboratories under the AI Singapore Programme (AISG Award No: AISG2-RP-2020-019); the RIE 2020 Advanced Manufacturing and Engineering (AME) Programmatic Fund (No. A20G8b0102), Singapore; the Shanghai University of Finance and Economics Research Fund (2021110066).

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
