# OpenReview forum: "Recurrent Temporal Revision Graph Networks"
_NeurIPS.cc/2023/Conference — NeurIPS 2023 poster_

### Official Review · Reviewer_2aAk · 2023-06-24

**Soundness:** 2 fair
**Presentation:** 3 good
**Contribution:** 3 good
**Rating:** 5
**Confidence:** 3

**Summary:**

Temporal graphs are more accurate for modeling real-world scenarios compared to static graphs, but the current approach of extending neighbor aggregation from static graphs to temporal graphs is computationally expensive when considering all historical neighbors. To address this, the authors propose a novel framework that uses recurrent neural networks with node-wise hidden states to integrate information from all historical neighbors for each node, ensuring complete neighbor information without subsampling biases.

**Strengths:**

1. The model proposed by the authors looks reasonable, at least  their experiments suggest so.

2. The authors do a good job in describing the model. They propose two measures of expressiveness for temporal graph models, specifically in the context of temporal link prediction and temporal graph isomorphism tests. They establish a theoretical connection between these measures and demonstrate the superior expressiveness of their framework, while also presenting new findings on the expressiveness of other baseline methods.

3. The ablation studies display the positive effect of each mechanism.

**Weaknesses:**

Some concerns in this paper should be discussed precisely.

1. In introduction, the heterogeneity in temporal revision is not explained.

2. Figure 1 lacks a specific description. The other baselines and the framework in this paper can be specified in the legend.

3. This paper lacks a description of Table 1. The authors can add a description of Table 1.

**Questions:**

1. Eq.3, 4 and 5 all refer to arbitrary learnable function, could you please specify what function is used in this paper?

2. In the experiment shown in Table 3, is it possible to display the results of all datasets?

**Limitations:**

The authors adequately addressed the limitations.

---

> ### Author Rebuttal · Authors · 2023-08-09
>
> Thank you for your constructive feedback which we will consider seriously in our revised paper.
>
> **Regarding your comment on weakness 1.** Thank you for this valuable feedback. We will add a simple explanation of what  heterogeneity means in the context of temporal revision. In simple terms, it means we will treat self-probing nodes differently in revision calculation.
>
> **Regarding your comment on weakness 2.** In our revised paper, we will refine the legend and caption to more clearly indicate which figure illustrates which.
>
> **Regarding your comment on weakness 3.** Thank you for this valuable feedback. We will provide a summary of the results and the main insights therein in the caption of Table 1.
>
> **Regarding your question 1.** We would like to refer you to Appendix A.2, where we have provided detailed implementations of all learnable functions we mentioned in the main paper. We will mention the implementation ideas in the main paper and refer readers to the Appendix for more details.
>
> **Regarding your question 2.** We selected 4 out of 8 datasets, due to limited computational resources, as representative which have covered different characteristics such as bipartite/non-bipartite and with/without edge features. We now have the full experiment results on all 8 datasets, which are provided in the uploaded PDF in the global rebuttal. As we can observe, the basic trend remains the same as before. Baseline methods show limited improvement when transitioning from the 1-layer to the 2-layer setting. By contrast, our proposed RTRGN exhibits significant improvements in the 2-layer setting (except for Wikipedia and Reddit where the precision of the 1-layer model is already as high as 98\% and the 2-layer model may suffer from overfitting) and clearly outperforms all baselines.
>
> We sincerely appreciate your valuable feedback and hope that our responses could adequately address your concerns.

---

> > ### Comment · Reviewer_2aAk · 2023-08-15
> > **Thanks a lot!**
> >
> > Thanks for your response and the additional experiments. I would like to keep my score.

---

### Official Review · Reviewer_UPBh · 2023-07-07

**Soundness:** 3 good
**Presentation:** 3 good
**Contribution:** 3 good
**Rating:** 5
**Confidence:** 5

**Summary:**

The paper studies the problem of temporal graph learning. The authors propose the recurrent temporal revision (RTR) layer, which involves learning a hidden state for each node and utilizing recursive neural networks to integrate information from all neighbors. Additionally, the authors introduce heterogeneity in RTR and theoretically demonstrate that this heterogeneous revision enhances expressiveness beyond Temporal-1WL.

**Strengths:**

S1. The majority of the paper is easy to follow.

S2. Experimental results show that RTR outperforms state-of-the-art methods.


**Weaknesses:**

W1. My main concern is with the motivation of paper. Why do we need to make each node embrace complete neighbor information? In real-world recommendation system data, the browsing/purchasing tendencies of users are often more closely related to their short-term interests. The inclusion of excessive and outdated information may introduce more noise to the model instead.

W2. The proposed RTR can be understood as weight learning and important neighbor selection based on complete historical information. Could changing the sampling strategy from selecting k nearest neighbors to selecting k most important historical neighbors for methods like TGN also result in similar performance improvements?

W3. The paper lacks an analysis of time complexity as well as a comparison of the model's parameter count.

Minor comments

M1. Page 1, Line 16: social network, recommender system -> social networks, recommender systems

M2. Page 4, Line 160: an learnable function -> a learnable function

M3. Page 8, Figure 4 caption: TGN and GRU-GCN fails to -> TGN and GRU-GCN fail to


**Questions:**

Please refer to W1-W3.

**Limitations:**

I have some concerns about the scalability of the proposed method.

---

> ### Author Rebuttal · Authors · 2023-08-09
>
> Thank you for your insightful feedback which we will consider seriously in our revised paper.
>
> **Regarding your comment W1.** We would like to emphasize that long-term user behavior has been a major focus in recommender systems. Numerous studies, such as [R1, R2, R3, R4, R5], have demonstrated its effectiveness in improving the accuracy of user behavior modeling. While it is true that short-term behaviors can have a significant impact on a user's near-future actions, they come with limited volume and thus may not fully capture the preferences of users over time, which may impair the performance in more general scenarios. The hidden state in our proposed RTRGN will be fed with the entire interaction sequence under a recurrent update scheme, which enables it to flexibly focus on either long-term or short-term behaviors based on the characteristic of the problems/datasets at hand. We will discuss more about the long-term and short-term trade-off and further explain the motivation behind our approach in the revised paper.
>
> **Regarding your question in W2.** Switching from k nearest neighbors to k most important historical neighbors could potentially benefit methods like TGN. However, 1) a fixed k can limit its performance. By contrast, RTRGN with its RNN and hidden state could potentially take a broader range of historical information into account; 2) accurately and effectively calculating/maintaining the k most important neighbors of dynamically evolving nodes can be time-consuming and challenging. In comparison, RTRGN with RNN can use simple nearest sampling to efficiently model the entire interaction history.
>
> **Regarding your question W3 and your concern about the scalability.** We would like to note that we have a thorough theoretical analysis of both the time and space complexity in Appendix A.4. Furthermore, we have practical run-time comparisons on real-world datasets, which can be found in Appendix D.6. To ensure easy access to this information, we will highlight the references to Appendix A.4 and D.6 in the revised paper. We will also include a comparison of the parameter counts of models. It is important to note that a substantial portion ($>$90\%) of the parameters in temporal graph models are the base embeddings (the node-wise hidden states of the base RNN, which are non-trainable, updated by the base RNN). An RTR layer has roughly 75\% more parameters than a TGN layer of the same configuration, where the increment of trainable parameters is mainly attributed to the additional state transition functions (Eq. (5)) and message function (Eq. (4)) introduced in RTRGN. When considering the whole model, further including the parameters of the base GRU and the time encoders, the growth ratio of trainable parameters of RTRGN compared with TGN would be less. As an example, when the hidden state dimension is 172, a 2-layer RTRGN model has about 2M trainable parameters, while a classic 2-layer TGN model with the same setting contains 1.57M trainable parameters.
>
> We sincerely appreciate your valuable feedback and hope our responses could largely address your concerns.
>
>
> [R1] User behavior retrieval for click-through rate prediction, SIGIR 2020.
>
> [R2] Search-based user interest modeling with lifelong sequential behavior data for click-through rate prediction, CIKM 2020.
>
> [R3] Adversarial filtering modeling on long-term user behavior sequences for click-through rate prediction, SIGIR 2022.
>
> [R4] Sampling is all you need on modeling long-term user behaviors for CTR prediction, CIKM 2022.
>
> [R5] TWIN: TWo-stage Interest Network for Lifelong User Behavior Modeling in CTR Prediction at Kuaishou, arXiv 2023.

---

### Official Review · Reviewer_sEkX · 2023-07-08

**Soundness:** 3 good
**Presentation:** 3 good
**Contribution:** 3 good
**Rating:** 7
**Confidence:** 4

**Summary:**

In this paper, the authors propose a novel aggregation layer based on a recurrent neural network dubbed recurrent temporal revision (RTR) for temporal graph networks to solve the dynamic temporal graph embedding problem. Specifically, a new aggregation function is proposed, which encodes neighbor state information and event information. This paper also proposes two expressive measures and uses them to demonstrate theoretically the difference from previous work. Empirical results on several well-known datasets show the effectiveness of the proposed method.

**Strengths:**

1．	The explanations of the components of the approach are clear and detailed.
2．	This work uses the proposed two measurements, and theoretically, fully justifies the difference from the previous methods. It also includes algorithm time complexity and running time analysis
3．	Ablation settings are very detailed.


**Weaknesses:**

1.	It is unclear which part of the ablation experiment corresponds to which of the two proposed measurements. It would be good to clarify that.
2.	It’s better to have an Algorithm line by line to clearly correspond to specific formulae and modules for Figure 1b.


**Questions:**

1.	Some symbols and formulations need to be clarified:
    a)	Eq. (4) should be accurately formulated.
    b)	r_t^0(t) in Line 148 should be a vector.
2.	Which variance or function in Eq3 and Eq4 can reflect the two proposed measurements?
3.	The organization of Section 3 should be improved. Each subsection ends abruptly without a clear takeaway message.
4.	Table 1 is hard to read.
5.	The differentiation between the current work and previous works (especially TGAT, and TGS) need to be explained more clearly.
6.	Why do the 2-layer experiments only run in selected datasets? What’s the selection criterion?

---

> ### Author Rebuttal · Authors · 2023-08-09
>
> Thank you for your constructive feedback which we will consider seriously in our revised paper.
>
> **Regarding your confusion about the correspondence between the ablation experiments and the two proposed measurements.** We would like to clarify that while the first two experiment sections (5.1 and 5.2) are dedicated to studying the expressiveness of temporal link prediction and temporal graph isomorphism test respectively, the entire ablation experiment section (5.3) was focused on the temporal link prediction setting: all the ablation experiments therein are conducted on the temporal link prediction task and measured by the average precision of temporal link prediction. Note that our theoretical analysis has suggested that the heterogeneous revision (HR) is the essence of improved expressiveness in the temporal graph isomorphism test, which is also verified by the results in Table 4 (w/o HR). This is the main reason why our ablation studies were primarily focused on temporal link prediction. Another reason is that most real-world problems revolve around link prediction and most datasets are also predominantly suited for the link prediction task.
>
> **Regarding your suggestion to enhance the illustration with the help of algorithm pseudocode.** We appreciate this constructive feedback, and will provide such an algorithm alongside and make necessary improvements to Figure 1b in our revised paper.
>
> **Regarding your question 1.** For a more accurately formulated Eq. (4), we would like to refer you to Appendix A.2, where we have provided detailed implementations of all learnable functions we mentioned in the main paper. We intended to present our framework from a general perspective in the main paper, deliberately separated the implementation details to the appendix, to encourage the exploration of potentially improved implementations in the future. Considering the importance of implementation details to a full understanding as suggested, we will mention the implementation ideas in the main paper and refer readers to the Appendix for more details. And yes, $r_t^0(t)$ in Line 148 should be a vector. We will enhance the notation, as well as other related ones, to make it clear.
>
> **Regarding your question 2.** Strictly speaking, the proposed expressiveness measurements are general and independent of the proposed model. We assume you mean which formula of the proposed model contributes to the improved expressiveness. In fact, the heterogeneous revision (Eq. (6)) is the key to the theoretical improvement of expressiveness, when assuming the full neighborhoods are involved in aggregation -- which is the common practice in theoretical analysis. While practically, as baseline methods typically only involve subsampled neighbors, our proposed Recurrent Temporal Revision framework (Eq. (3) - (5)) contributes to the majority of practical improvements of performance, due to its better modeling of neighborhood information.
>
> **Regarding your question 3.** Thank you for this valuable feedback. We will add takeaway messages to each subsection.
>
> **Regarding your question 4.** We will briefly describe the main message of Table 1 in its caption. We will also try to improve its coloring and layout for better readability.
>
> **Regarding your question 5.** We assume by TGS, you were referring to TGN. Despite other minor differences, the most important distinction between our proposed model and baseline models like TGN and TGAT lies in the aggregation strategy. We adopt the proposed Recurrent Temporal Revision, while TGN and TGAT employ the classic temporal aggregation. As illustrated in Figure 1a, classic temporal aggregation typically involves subsampled neighbors in each step, which leads to incomplete and biased neighbor information. By contrast, the proposed RTRGN, ideally, can integrate the whole information of neighbors. We will clarify this in the revised paper.
>
> **Regarding your question 6.** We selected 4 out of 8 datasets, due to limited computational resources, as representative which have covered different characteristics such as bipartite/non-bipartite and with/without edge features. We now have the full experiment results on all 8 datasets, which are provided in the uploaded PDF in the global rebuttal. As we can observe, the basic trend remains the same as before. Baseline methods show limited improvement when transitioning from the 1-layer setting to the 2-layer. By contrast, our proposed RTRGN exhibits significant improvements in the 2-layer setting (except for Wikipedia and Reddit where the precision of the 1-layer model is already as high as 98\% and the 2-layer model may suffer from overfitting) and clearly outperforms all baselines.

---

### Official Review · Reviewer_gXqR · 2023-07-08

**Soundness:** 3 good
**Presentation:** 3 good
**Contribution:** 4 excellent
**Rating:** 7
**Confidence:** 3

**Summary:**

The paper introduces a novel framework called Recurrent Temporal Revision (RTR) to address the challenge of neighbor subsampling in temporal graphs, an issue commonly encountered in real-world applications like social networking, recommender systems, traffic forecasting, and crime analysis. RTR serves as a standard building block for any temporal graph network, utilizing a recurrent neural network to integrate all neighbor information through a hidden node state. This hidden state mitigates the problem caused by neighbor subsampling and aims to capture complete neighbor information.

The paper also introduces a new concept of "temporal revision" to update the hidden state by capturing the state changes of neighbors. To further boost the theoretical expressiveness of this technique, the authors propose incorporating heterogeneity into temporal revision by recursively identifying and marking certain nodes as specialties in the revision calculation process.

The authors suggest that this new approach offers superior expressiveness in terms of temporal link prediction and temporal graph isomorphism test, a claim that is backed by theoretical proofs and experimental results. Additionally, they provide a new Ecommerce dataset for evaluating temporal graph models in real-world settings. Their experimental results indicate the proposed framework's significant improvement over state-of-the-art methods, particularly as the number of aggregation layers increases.

**Strengths:**

- The authors introduce innovative ways of evaluating the expressiveness of graph algorithms by defining two novel metrics: the temporal graph isomorphism test and the temporal graph link prediction. They apply these new measures to several state-of-the-art models, successfully establishing an equivalence between Temporal-1WL and these baseline models which implies the shared drawbacks of these models.
- The paper presents a unique temporal learning model grounded on two inventive ideas: 1) Utilizing hidden states to retain aggregated information which is subsequently updated through a temporal revision layer, and 2) Embedding heterogeneity by giving special treatment to the central node during the recursive revisioning process. The authors compellingly argue that their proposed method outperforms state-of-the-art models in terms of expressiveness, as defined within the context of this paper.
- The proposed method demonstrates promising performance in real-world temporal graph tasks, particularly in the temporal link prediction task. Additionally, it excels in the graph isomorphism test examples, reinforcing its potential for practical applications in various fields.

**Weaknesses:**

- The rationale behind the new definition of expressiveness isn't effectively conveyed in the paper. While it's true that there isn't a universally accepted definition for the expressiveness of a temporal graph model at present, the paper falls short in clarifying why its proposed isomorphism test improves upon existing ones. The lack of a clear explanation leaves a gap in understanding the superiority of the proposed metrics.

**Questions:**

From the proof of proposition 3, the extra expressiveness of the proposed model comes from the heterogenous revision which could be in principle added to other IMP-TGN representation model. Can we conclude that heterogeneity help improve expressiveness of temporal graph model?

---

> ### Author Rebuttal · Authors · 2023-08-09
>
> Thank you for your insightful feedback which we will consider seriously in our revised paper.
>
> **Regarding your consideration of the rationale behind our new definition of expressiveness.** We develop the set of new expressiveness measures mainly for the following two reasons. First, as you have reckoned, there is currently no universally accepted definition for the expressiveness of a temporal graph model, especially for temporal link prediction. Our definition is an effort towards addressing this gap, providing a consistent definition for the expressiveness of both the temporal graph isomorphism test and the temporal link prediction. Second, existing definitions of expressiveness are relatively complex. By contrast, our proposed ones are concise and straightforward, avoiding the introduction of Identifiable Set as in [5] and Temporal Computation Tree as in [33]. We will reorganize related text to make the rationale behind more clear.
>
> **Regarding your question about heterogeneity.** Indeed, previous studies [43], as mentioned in our related work, have shown that incorporating heterogeneity is a principal approach to enhancing the expressiveness of static graph models. Our work in a sense has extended this finding to temporal graph models. We believe that it could potentially also lead to improvements in expressiveness if integrated into other IMP-TGN models. However, it requires further verification.

---

> > ### Comment · Reviewer_sEkX · 2023-08-21
> > **Thanks for rebuttal.**
> >
> > Hi authors, Thanks for your rebuttal that solved my concerns. I will retain my score.

---

### Author Rebuttal · Authors · 2023-08-09

We sincerely thank all the reviewers for their constructive feedback. We provide the extended Table 3 in the attached PDF, while all other concerns reviewers raised are addressed in their respective rebuttals.

---

### Decision · Program_Chairs · 2023-09-21

**Decision:**

Accept (poster)

**Comment:**

This work proposes a novel aggregation layer based on a recurrent neural network dubbed recurrent temporal revision (RTR) for temporal graph networks. Specifically, the new aggregation function encodes neighbor state information and event information. This paper also proposes two expressive measures and uses them to demonstrate theoretically the difference from previous work.

This paper provides a substantial improvement in our understanding of temporal graph representation learning. After reading the paper in depth, I found the analysis in the paper both well-written and insightful (the Oscillating CSL example is very good. I believe Proposition 2 is incorrect but that does not detract from the rest of the paper). The reviews and rebuttals further solidify this is a good paper.

Proposition 2 seems incorrect: The error in the proof is to assume that Time-then-graph in [5] proses to map the set of edges. Time-then-graph [5] works in the Oscillating CSL since the authors propose encoding the edges as a sequence "edge attributes are sequences of non-existing (0) and existing (1) over time", not sets of edges. The time-then-graph of GRU-GCN results in Table 4 are probably due to an implementation choice of the authors, not a limitation of Time-then-graph. We trust the authors will address this misunderstanding in the camera ready. This does not detract from the paper.